# Allosteric substrate release by a sialic acid TRAP transporter substrate binding protein
Niels Schneberger [1], Philipp Hendricks [1], Martin F. Peter [1,4], Erik Gehrke[1], Sophie C. Binder [1], Paul-Albert Koenig[2], Stephan Menzel[2], Gavin H. Thomas [3] & Gregor Hagelueken [1] ✉

The tripartite ATP-independent periplasmic (TRAP) transporters enable *Vibrio cholerae* and *Haemophilus influenzae* to acquire sialic acid, aiding their colonization of human hosts. This process depends on SiaP, a substrate-binding protein (SBP) that captures and delivers sialic acid to the transporter. We identified 11 nanobodies that bind specifically to the SiaP proteins from *H. influenzae* (HiSiaP) and *V. cholerae* (VcSiaP). Two nanobodies inhibited sialic acid binding. Detailed structural and biophysical studies of one nanobody-SBP complex revealed an allosteric inhibition mechanism, preventing ligand binding and releasing pre-bound sialic acid. A hydrophobic surface pocket of the SBP is crucial for the allosteric mechanism and for the conformational rearrangement that occurs upon binding of sialic acid to the SBP. Our findings provide new clues regarding the mechanism of TRAP transporters, as well as potential starting points for novel drug design approaches to starve these human pathogens of important host-derived molecules.

Substrate binding proteins (SBPs) play an important role in many transport processes that are catalyzed by either ATP-binding cassette (ABC) transporters, tripartite tricarboxylate transporters (TTTs) or tripartite ATP-independent periplasmic (TRAP) transporters[1,2]. The SBPs are either freely diffusing in the periplasm of Gram-negative bacteria or are anchored to the outer surface of the membrane of Gram-positive bacteria. Their role is to scavenge nutrients and deliver them to their cognate transporter. Simulations have shown that under conditions where the substrate is scarce, SBPs increase the efficiency of transport by acting as substrate sponges[3]. Their importance is vividly demonstrated by bacterial genomes that encode hundreds of different SBPs, such as *Rhodoplanes sp.* Z2-YC6860[4].

SBPs have been studied in great detail over the past decades. High-resolution crystal structures, biophysical methods or molecular dynamics simulations have provided many important insights into their structure and function. In most cases, the substrate binding mechanism involves large scale conformational changes that are often compared to a Venus flytrap, with the protein tightly wrapping around the substrate. In the apo state (i.e. without substrate), substrate binding proteins behave quite differently, with some resting in a relatively stable open state, while others dynamically flip between their open and closed states[1,5–14].

The fact that SBP-dependent TRAP transporters are not present in eukaryotes but play a role in pathogenesis and host colonization, makes them interesting from a drug discovery point of view. Inspired by our previous finding that the SBPs of TRAP transporters can bind to and even (partially) close around artificial peptides[15], albeit with low affinity, we aimed to isolate VHH antibodies[16] (variable domain of the H-chain of heavy chain only antibodies, also known as Nanobodies™) that would hopefully inhibit the function of the SBP, for instance by blocking its substrate binding site.

Here we present a set of 11 VHH antibodies that were raised by immunizing two different alpacas with either HiSiaP or VcSiaP, the SBPs (also known as "P-domains") from the sialic acid TRAP transporters HiSiaPQM from *Haemophilus influenzae* and VcSiaPQM from *Vibrio cholerae*. These transporters are known to be important for the virulence of these two human pathogens by scavenging host-derived sialic acid and either incorporating it into the bacterial cell envelope to allow immune evasion or in providing a rich source of carbon and nitrogen for the bacterium[17–19]. Due to the importance of these systems in bacterial colonization they have become model TRAP transporters and their mechanisms are actively studied at the functional and structural levels[6,9,10,15,20–27].

The relatively strong interaction of sialic acid (Neu5Ac) with SiaP is the first step in the transport cycle, after which the ligand-bound SBP is recognized by the SiaQM protein in the bacterial inner membrane[20]. The transporter uses electrochemical gradients across the membrane to drive the

[1]Institute of Structural Biology, University of Bonn, Venusberg-Campus 1, 53127 Bonn, Germany. [2]Core Facility Nanobodies, University of Bonn, Venusberg-Campus 1, 53127 Bonn, Germany. [3]Department of Biology (Area 10), University of York, York, YO10 5YW, UK. [4]Present address: Biochemistry Center, Heidelberg University, Im Neuenheimer Feld 328, 69120 Heidelberg, Germany. ✉e-mail: hagelueken@uni-bonn.de

release of sialic acid from SiaP and subsequent transport across the membrane[26]. To learn more about how the VHHs might influence the first step in the transport cycle we used biophysical methods such as size exclusion chromatography (SEC) combined with multi-angle light scattering (MALS) and isothermal titration calorimetry (ITC) to thoroughly characterize the VHHs and their interactions with SiaP. Two of the nanobodies were found to strongly inhibit Neu5Ac binding and we used X-ray crystallography to determine the structure of VcSiaP in complex with a nanobody that allosterically inhibits sialic acid binding to this SBP. Using site-directed mutagenesis, ITC and further crystal structures, we dissected the VcSiaP/VHH interface and were able to identify the molecular reason for the allosteric effect. Our results shed new light on the structural mechanism behind the substrate-induced closure of TRAP transporter SBPs.

## Results

### Selection of VHHs against HiSiaP and VcSiaP

Two alpacas were immunized with either VcSiaP or HiSiaP, to obtain VHH antibodies. Based on a phylogenetic analysis and ELISA screening, 11 VHHs (2 for VcSiaP and 9 for HiSiaP) were picked for further analysis (Fig. 1a). As

expected, all eleven proteins share a highly conserved backbone sequence and have variable complementary determining regions (CDRs). Their overall sequence identities varied from 65% to 81% (Fig. 1b). There was no clear sequence motif that distinguished the VHHs raised against the two antigens (i. e. VcSiaP or HiSiaP) (Fig. 1b, c).

The two nanobodies targeting VcSiaP (VHH$_{VcP}$ #1, VHH$_{VcP}$ #2) could be expressed and purified with high yield and purity. We used size exclusion chromatography combined with multi-angle light scattering (SEC-MALS) experiments as a quick test for binding of the two VHHs to the target protein. Both formed a stable complex with VcSiaP (Fig. 2a). For the individual proteins and both 1:1 complexes, the experimental molecular weights (MWs) were in good agreement with theoretical values (Fig. 2a). A quantitative binding analysis by ITC revealed a dissociation constant ($K_D$) of 162 nM for the VcSiaP/VHH$_{VcP}$ #1 complex and a $K_D$ of 13 nM for the VcSiaP/VHH$_{VcP}$ #2 complex (Supplementary Fig. 1). While the VcSiaP/VHH$_{VcP}$ #2 binding reaction had a relatively large enthalpic contribution ($\Delta H$), the VcSiaP/VHH$_{VcP}$ #1 binding reaction showed a much smaller $\Delta H$ and a correspondingly large term for $\Delta S$, indicating a mostly entropy-driven reaction (Supplementary Fig. 1a, b).

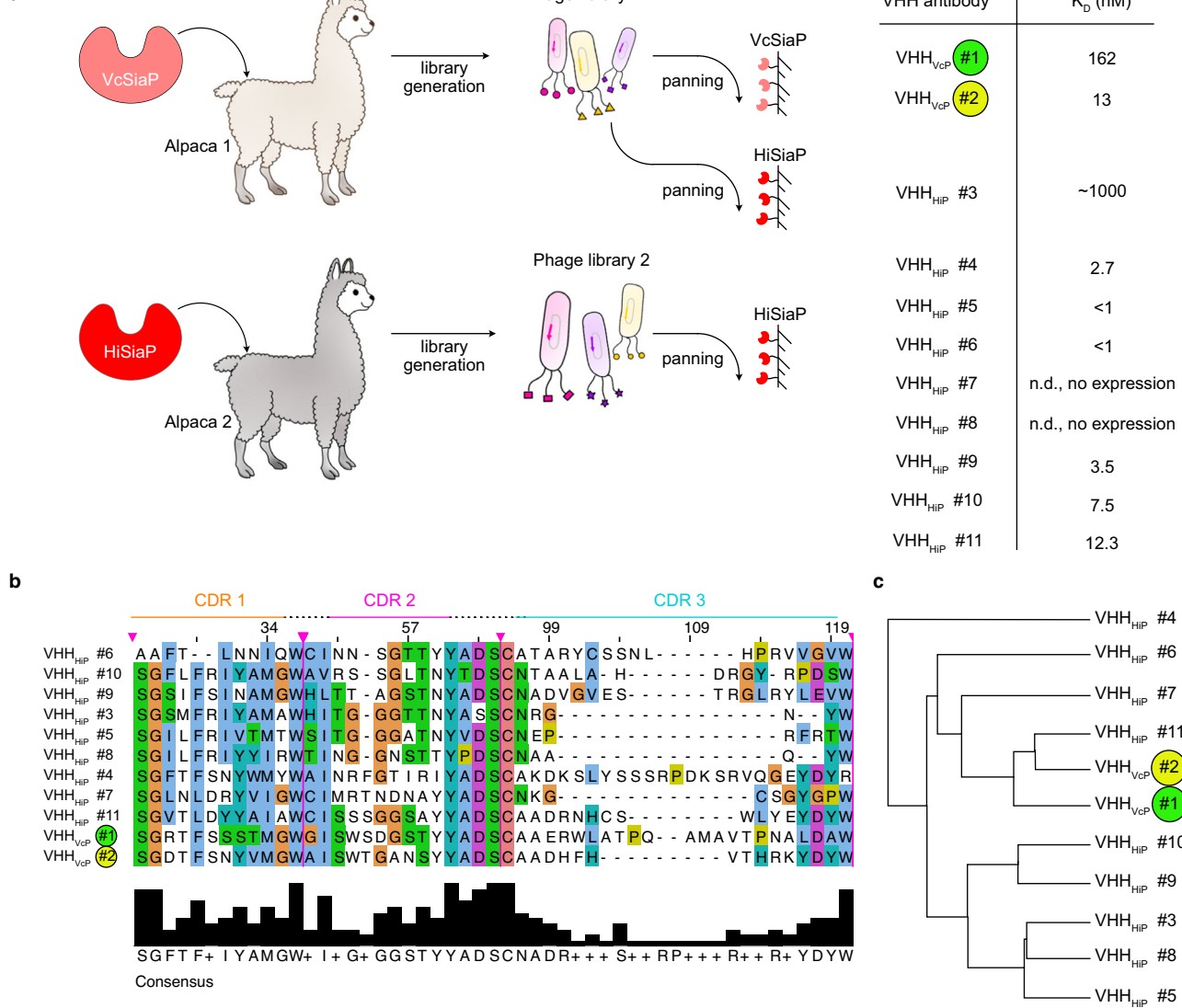

**Fig. 1 | VHH antibodies raised against VcSiaP and HiSiaP. a** Two alpacas were immunized with either VcSiaP or HiSiaP, resulting in VHH libraries 1 and 2. Three panning experiments with immobilized VcSiaP or HiSiaP resulted in 11 VHH antibodies that specifically bound VcSiaP (Vc#1 and Vc#2) or HiSiaP (Hi#3-Hi#11). VHH$_{VcP}$ #1/2 are color coded, since they are the main focus of the following experiments. The $K_D$ values were determined by ITC. **b** A Clustal Omega alignment[44] of the CDR of the VHHs shown in **a**). **c** Cladogram of the VHH antibodies based on the alignment in panel **b**).

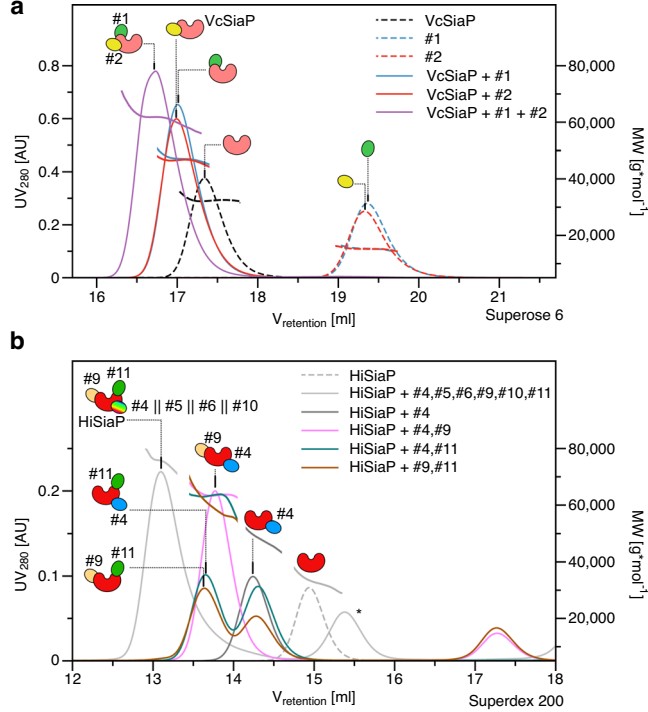

**Fig. 2 | Epitope binning. a** A set of SEC-MALS experiments showing that VHH$_{VcP}$ #1 and #2 can simultaneously bind to VcSiaP, forming a ternary complex. **b** A set of SEC-MALS experiments showing that at most three different VHHs bind to HiSiaP. Together with the data shown in Supplementary Fig. 2, these experiments revealed that VHH$_{HiP}$ #9 and VHH$_{HiP}$ #11 bind to separate epitopes, while VHH$_{HiP}$ #4, 5, 6, 10 share an overlapping epitope. The epitope binning experiment was performed once (n = 1). The asterisk denotes a peak that resulted from VHH$_{HiP}$#8, which was excluded from further analyses because it did not bind to HiSiaP.

Of the nine HiSiaP-binding VHHs (VHH$_{HiP}$ #3-11), two could not be expressed and purified in sufficient amounts, namely VHH$_{HiP}$ #7, 8. All other VHHs were "well behaved", albeit with some variations in the expression yield. Again, SEC-MALS experiments provided an initial indication for the binding to the target protein and in each case, a shift of the HiSiaP peak to smaller retention volumes was observed (Supplementary Fig. 2a, b). All complexes showed a good correlation between their experimental and calculated MWs (Supplementary Table 1). The shift for the HiSiaP/VHH$_{HiP}$ #3 complex was however smaller than expected (Supplementary Fig. 2d), fitting to its relatively low affinity (K$_D$ = 0.89 μM[21]). Interestingly, this VHH was found by a different panning approach, using the nanobody library raised against VcSiaP and the structurally similar HiSiaP as a bait protein (AA sequence identity: 50.17% (without signal peptide)) (Fig. 1a).

### Epitope binning of VHHs
We wondered, whether the different nanobodies would bind to different epitopes of the two SBPs. Since we only had two VHHs for VcSiaP, it was straight forward to investigate this by ITC and we found that an equimolar mixture of VcSiaP and VHH$_{VcP}$ #1 was still able to bind VHH$_{VcP}$ #2, with only minor changes in all binding parameters compared to the titration of VHH$_{VcP}$ #2 to VcSiaP alone (Supplementary Fig. 3a). Furthermore, dynamic light scattering (DLS) experiments (Supplementary Fig. 3b), SEC-MALS analysis (Fig. 2a) and thermal stabilization assays (Supplementary Fig. 3c) supported the existence of the tripartite VcSiaP/VHH$_{VcP}$ #1, 2 complex. While the proteins alone gave rise to significantly smaller hydrodynamic radii than those of the dimeric complexes, the heterotrimeric complex had a hydrodynamic radius of 3.8 nm, slightly larger than the calculated one (3.4 nm, Supplementary Fig. 3b). Interestingly, the thermal stability of VcSiaP was drastically increased by about 16.5 °C when both VHHs were bound in a 1:1:1 manner, while a stabilization of only about 6 °C

was observed for the 1:1 complexes (Supplementary Fig. 3c). We concluded that VHH$_{VcP}$ #1, 2 can independently bind to VcSiaP at different epitopes.

For an epitope binning of the HiSiaP-binding VHHs, we performed a SEC-MALS analysis. Firstly, to narrow down the large amounts of possibilities, we injected a mixture of VHH$_{HiP}$ #4, 5, 6, 9, 10, 11 and HiSiaP to estimate the largest possible complex. The resulting SEC peak had an elution volume of ~13.1 ml on a Superdex 200 10/300 column and an experimental MW of 75 kDa, indicating three VHHs (each ~15 kDa) per HiSiaP ( ~ 30 kDa) (Fig. 2b). A systematic analysis of all 15 possible combinations of two different VHH antibodies and HiSiaP, suggested unique epitopes for VHH$_{HiP}$ #11 and VHH$_{HiP}$ #9, while the remaining nanobodies, VHH$_{HiP}$ #4,5,6,10, appeared to bind to a third epitope or at least to overlapping epitopes (Supplementary Fig. 2). Thus, the heterotetrameric complex in the initial experiment was most likely built up from HiSiaP bound to VHH$_{HiP}$ #9,11 and always one of the remaining VHHs. The gel filtration experiments in Fig. 2b support this conjecture. Note that due to the large number of combinations, not all possible variations of HiSiaP with three different VHHs were tested.

### VHHs inhibit Neu5Ac binding by VcSiaP and HiSiaP
We thought it likely that some of the VHHs might influence the conformational switching that TRAP transporter SBPs undergo during substrate binding[1,9,10,20]. To analyze this, we performed two sets of ITC experiments. In the first set, SiaP was initially loaded with sialic acid (e.g. "VcSiaP[Neu5Ac]") and then titrated with one of the VHHs in two separate ITC runs. In the second set of experiments, SiaP was first incubated with one of the VHHs (e.g. VcSiaP[VHH$_{VcP}$ #2]) and then titrated with sialic acid.

In most cases, the binding of VHH and sialic acid was independent of each other. However, VHH$_{VcP}$ #2 and VHH$_{HiP}$ #11 showed interesting effects, which were further analyzed by a sequential ITC experiment as described for VcSiaP and VHH$_{VcP}$ #2 in the following section. The first titration in the series, VcSiaP vs Neu5Ac, had the same result as observed before – clear binding of the Neu5Ac molecule to VcSiaP[7]. The molar ratio was determined to ~1 thus, the amount of VcSiaP in the ITC cell was saturated with Neu5Ac (Fig. 3a). For the second binding reaction of VHH$_{VcP}$ #2 to VcSiaP[Neu5Ac] we observed an endothermic reaction (Fig. 3b), whereas the binding of either sialic acid or VHH$_{VcP}$ #2 alone to VcSiaP was clearly exothermic (Fig. 3a, Supplementary Fig. 1b). Also here, the titration was complete at a molar ratio of ~1.

A possible explanation for this observation would be a competitive binding reaction, where the stronger binding VHH (K$_D$ = 16 nM) outcompetes the weaker binding Neu5Ac (K$_D$ ~ 200 nM). Because the enthalpic contribution of the Neu5Ac binding reaction is larger than that of VHH$_{VcP}$ #2, a net endothermic reaction is observed. Indeed, the data could be fitted with a competitive binding model[28] and the resulting binding parameters (Supplementary Fig. 4, Supplementary Table 2) were in agreement with the single binding experiments described above. Next, HiSiaP (which does not bind VHH$_{VcP}$ #2, Supplementary Fig. 5b) was titrated to the mixture from the second titration to measure the amount of the now freely diffusing sialic acid in the ITC cell. Indeed, a molar ratio of ~2 for this HiSiaP vs Neu5Ac titration indicated, that all sialic acid added in titration #1 was available for binding HiSiaP (Fig. 3c). Thus, VHH$_{VcP}$ #2 is not only able to inhibit binding of Neu5Ac towards VcSiaP but can effectively outcompete binding of sialic acid.

Consistent with this explanation, Neu5Ac did not bind to VcSiaP[VHH$_{VcP}$ #2]. Furthermore, a sub-stoichiometric preincubation of VcSiaP with the VHH (VcSiaP[VHH$_{VcP}$ #2] 1:0.5) resulted in only half of the VcSiaP molecules binding to Neu5Ac (Supplementary Fig 5).

Similar effects were observed for VHH$_{HiP}$ #11. Also here, the VHH outcompeted Neu5Ac in ITC competition experiments and once bound to the VHH, the SBP was no longer able to bind to Neu5Ac (Supplementary Fig. 6).

### Structural analysis of VcSiaP targeting VHHs
In order to gain structural insights into the interesting properties of the different VHHs, we performed crystallization experiments. Single crystals of

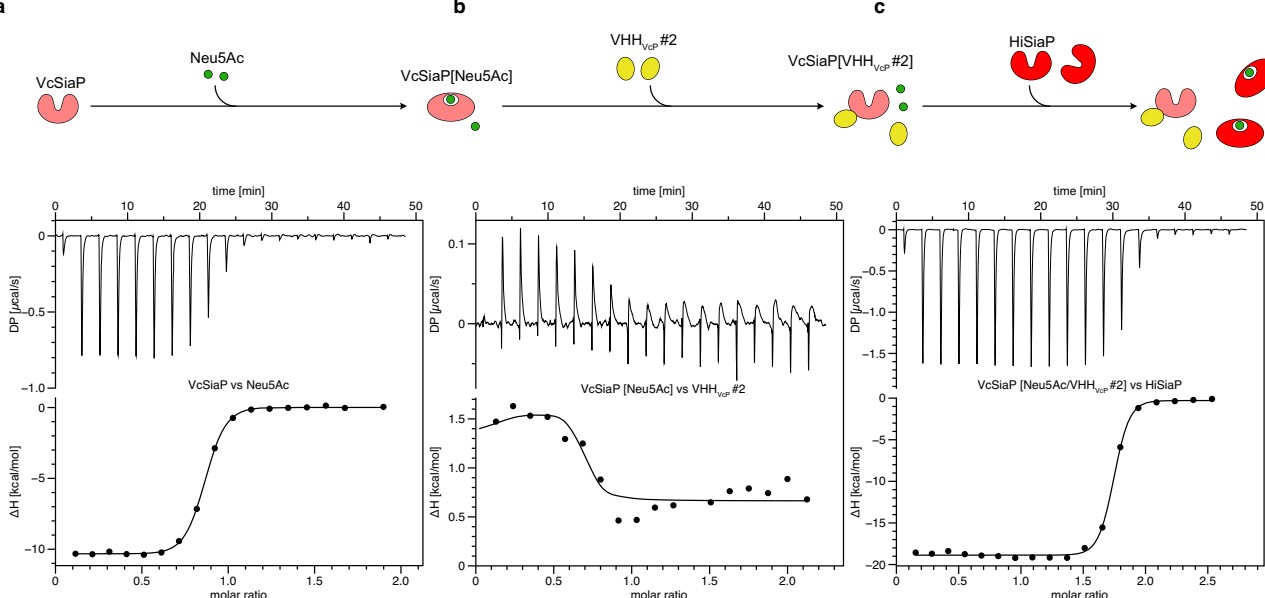

**Fig. 3 | VHH_VcP_#2 expels bound Neu5Ac from VcSiaP. a** Titration of VcSiaP vs Neu5Ac. **b** The mixture resulting from the experiment shown in panel **a)** was titrated against VHH_VcP_#2. **c** The mixture resulting from the experiment shown in panel **b)** was titrated against HiSiaP. For all titrations, the raw data are shown on the top

and the integrated binding heat vs the molar ratio on the bottom. The titration series was done twice (*n* = 2, technical replicate). The replicate of this set of experiments is shown in Supplementary Fig. 4.

## Table 1 | Data collection and refinement statistics

|  | VcSiaP_apo-VHH_VcP_#1 | VcSiaP_apo-VHH_VcP_#2 | VcSiaP W73A [Neu5Ac]-VHH_VcP_#2 |
|---|---|---|---|
| PDB identifier | **9FVC** | **9FVB** | **9FVE** |
| **Data collection** | | | |
| Space group | $P\,2_1\,2_1\,2$ | $C\,1\,2\,1$ | $C\,1\,2\,1$ |
| Cell dimensions | | | |
| *a, b, c* (Å) | 169.61, 72.37, 85.60 | 151.30, 50.34, 133.99 | 223.57 153.11 210.49 |
| α, β, γ (°) | 90, 90, 90 | 90, 114.9, 90 | 90 89.99 90 |
| Resolution (Å) | 47.18–2.644 (2.71–2.64) | 46.38 - 2.053 (2.126 - 2.053) | 33.8–2.81 (2.91–2.81) |
| $R_{merge}$ | 0.09713 (0.5301) | 0.09801 (0.7936) | 0.06463 (0.3428) |
| I / σI | 6.90 (1.34) | 7.48 (1.04) | 8.09 (2.41) |
| Completeness (%) | 99.61 (95.33) | 99.54 (95.99) | 99.05 (99.39) |
| Redundancy | 6.9 (6.9) | 3.5 (3.4) | 5.5 (3.7) |
| **Refinement** | | | |
| Resolution (Å) | 47.18–2.644 (2.71–2.64) | 46.38–2.053 (2.126–2.053) | 33.8–2.81 (2.91–2.81) |
| No. reflections | 31516 (3009) | 57530 (5499) | 170742 (17064) |
| $R_{work}$ / $R_{free}$ | 0.1947 / 0.2569 | 0.1952 / 0.2572 | 0.2296 / 0.2649 |
| No. atoms | 6701 | 7186 | 39979 |
| Protein | 6644 | 6647 | 39703 |
| Ligand/ion | 0 | 106 | 528 |
| Water | 171 | 493 | 0 |
| *B*-factors | 44.72 | 34.38 | 45.04 |
| Protein | 44.77 | 34.26 | 45.07 |
| Ligand/ion | - | 48.22 | 40.05 |
| Water | 38.79 | 34.66 | - |
| R.m.s. deviations | | | |
| Bond lengths (Å) | 0.013 | 0.013 | 0.018 |
| Bond angles (°) | 1.36 | 1.20 | 1.97 |

Values in parentheses are for highest-resolution shell.

the VcSiaP/VHH_VcP #1 & VcSiaP/VHH_VcP #2 apo complexes were obtained after incubation at 20 °C for 2 days. The crystal structures were solved by molecular replacement[29] at 2.6 Å (VcSiaP/VHH_VcP #1) and 2.05 Å (VcSiaP/VHH_VcP #2) using VcSiaP (PDB ID: 4mag[7]) and a BtuF specific VHH (PDB ID: 5ovw[30]) as search models (Table 1). For both structures, the electron density clearly showed all residues of VcSiaP (1-299), whereas the C-terminal HA-His₆ tags of the VHHs were disordered. Both VHHs bind to the N-lobe of VcSiaP at different concave surface regions (Figs. 4, 5).

The interaction interface of the VcSiaP/VHH_VcP #1 structure was analyzed using the PDBePISA online tool and amounts to an area of 556.5 Å² [31]. The most striking feature of this interface is the side chain W101 in CDR3 of VHH_VcP #1, which lies flat on the VcSiaP surface and forms hydrophobic interactions with residues A24, L37, and A38 of VcSiaP (Fig. 4b, c). In addition, VHH_VcP #1 interacts with VcSiaP by polar interactions involving residues R27 (E28 (3.6 Å)), D55 (Y40 (3.2 Å)), E99 (Y17 (2.9 Å), K21 (2.5 Å)), R100 (E28 (2.6, 2.9 Å)), W101 (D25 (2.8 Å), L39 (3.1 Å)), A103 (L39 (2.9 Å)), T104 (Q44 (2.4 Å)), T111 (V12 (2.7 Å), and D116 (K21 (2.9 Å)). Several hydrophobic interactions are also involved in the binding (Fig. 4b, c).

The VHH_VcP #2/VcSiaP structure has a slightly larger ( + 15%) inter-action area of 639.3 Å² as determined by the PDBePISA server[31]. The interface is centered around F101 on CDR 3 of the nanobody, which penetrates into a largely hydrophobic cluster at the surface of VcSiaP, formed by residues R49, Q53, W73, F112 and W114 (Fig. 5b, c). Also notable is cation-pi interaction between VcSiaP R49 and VHH_VcP #2 W53 with a distance of 3.6 Å. We identified a large number of further polar and ionic interactions summarized in Fig. 5c. Interestingly, the VHH_VcP #2/VcSiaP interface has fewer electro-static and polar interactions than found for VHH_VcP #1 despite the higher affinity (and drastically larger binding enthalpy).

### Allosteric inhibition by VHH_VcP #2 is achieved by F101 acting as a "door stop"

Considering the strong inhibitory effect of VHH_VcP #2 on Neu5Ac binding, it was surprising that the VHH does not bind in the Neu5Ac binding site of the SBP, indicating an allosteric mechanism of inhibition. Interestingly, VHH_VcP #2 binds close to a region of VcSiaP (residues 70–100) that undergoes a large number of relatively small-scale structural rearrangements between the open- and closed structures of the protein. This is evident from a difference distance

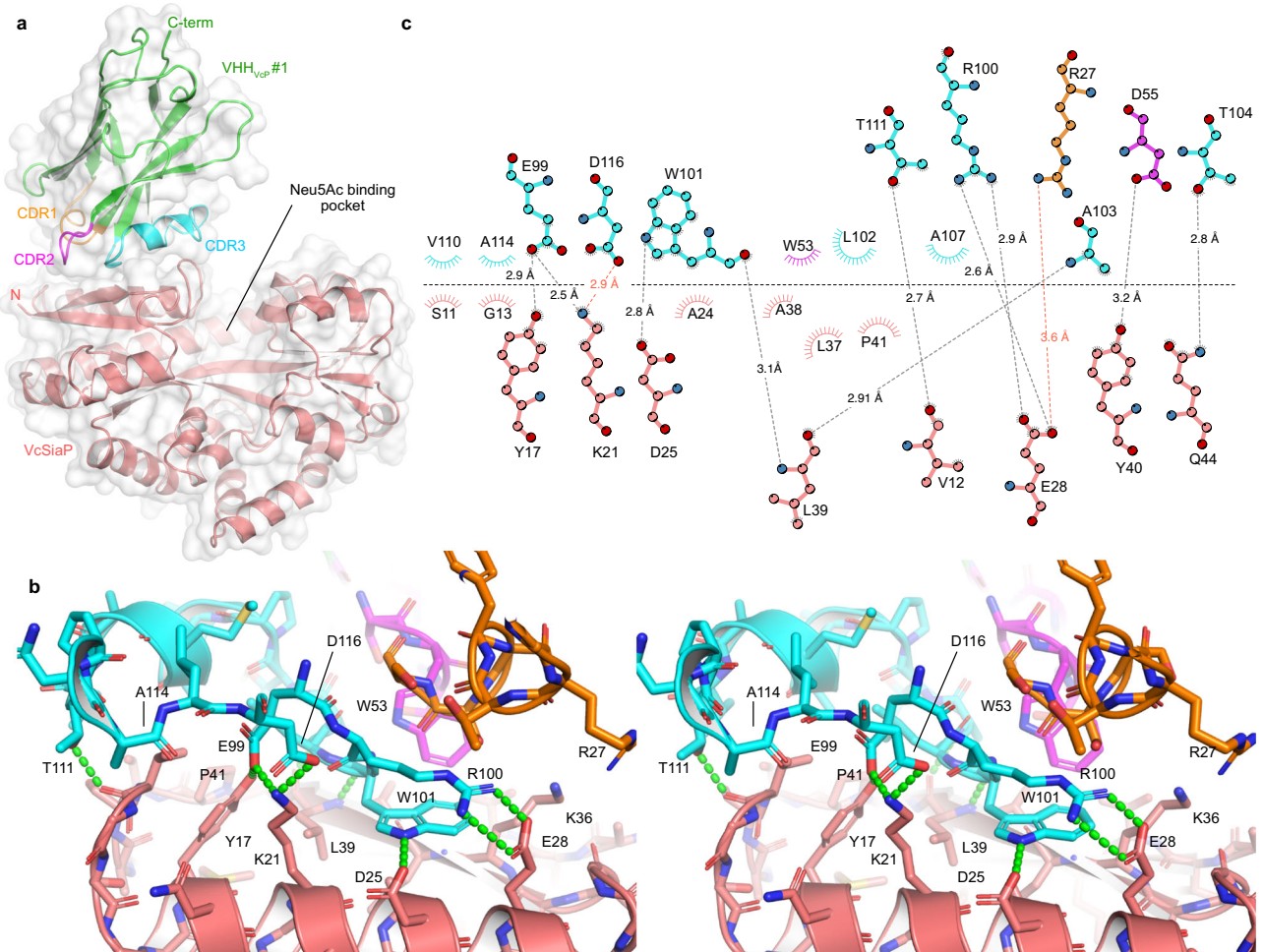

**Fig. 4 | Crystal structure of the VcSiaP/VHH_VcP #1 complex. a** Overall structure as a cartoon model. The CDR of VHH_VcP #1 are color coded throughout all panels. **b** Stereo-pair showing a close-up of the interaction. Polar interactions are indicated by green dashes. **c** Schematic representation (Ligplot+[45]) of the interacting residues (VHH_VcP #1 at the top, VcSiaP at the bottom). Polar interactions are shown as dashed lines and arcs represent van-der-Waals interactions.

matrix (DDM) between the two crystal structures that illustrates the distance change between each pair of residues during the open to closed transition (Fig. 6a, b).

In contrast, VHH_VcP #1 binds to a directly adjacent region (residues 1–50) that (as judged by the DDM) is almost perfectly rigid during the transition. Overall, the most striking difference between apo VcSiaP and our VHH_VcP #2 complex structure is the area surrounding W73 of VcSiaP, which is itself forced into a slightly different conformation and thereby creates the pocket that accommodates F101 of the nanobody (Fig. 6c).

To check, whether this rearrangement is indeed key to the allosteric inhibition, we mutated the tryptophane to an alanine, with the intention to create more space and allow the structural transition in the presence of VHH_VcP #2. Despite this rather drastic change to the binding epitope, the VHH still bound to the mutated protein with a very similar affinity ($K_D = 22.8$ nM for mutant vs 13 nM for WT) (Supplementary Fig. 8). In contrast, the F101A mutation of the VHH did not form a stable complex with VcSiaP (Supplementary Fig. 8).

Next, we titrated VcSiaP W73A[VHH_VcP #2] with Neu5Ac and we clearly observed binding in this ITC experiment (Fig. 6d, purple). The affinity ($K_D = 1.2$ μM) was however reduced compared to the wild-type SBP ($K_D = 211$ nM, Fig. 6d, green), resulting in a less steep binding isotherm. In turn, VHH_VcP #2 was able to bind to VcSiaP W73A [Neu5Ac] without the competitive binding effects that were observed for wild-type VcSiaP Supplementary Fig. 4). A sequential ITC experiment (as sketched in Fig. 3) showed that Neu5Ac remains bound to VcSiaP W73A when VHH_VcP #2 is titrated and binds to VcSiaP (Supplementary Fig. 4). Furthermore, we found that the

thermal stabilization of VcSiaP/VHH_VcP #2 by sialic acid was partially recovered by the W73A mutation while the wild type cannot be stabilized by sialic acid when bound to VHH_VcP #2 (Supplementary Fig. 9). This substrate induced stabilization was not affected by VHH_VcP #1. Very interestingly, the W73A mutant had a stronger affinity towards Neu5Ac ($K_D = 76.9$ nM) than the wild-type SBP ($K_D = 211$ nM). Of all the SiaP mutants we have characterized in recent years, this is a unique and very interesting observation.

## Crystal structure of VcSiaP W73A/VHH_VcP #2

To investigate the effect of the VcSiaP W73A mutant in detail, we crystallized it in complex with VHH_VcP #2 and Neu5Ac. A diffraction dataset at 2.8 Å resolution was collected and the structure was solved by molecular replacement using the closed-state VcSiaP without Neu5Ac (PDB-ID: 7a5q[15]) and VHH_VcP #2 as separate models (Table 1). The C2 unit cell contained 24 protein chains and thus 12 VcSiaP/VHH_VcP #2 complexes. Supplementary Fig. 10 shows an overlay of the individual complexes. For each of the 12 P-domains, strong difference electron density was observed at the sialic acid binding site. The difference density fitted sialic acid in all cases (Supplementary Fig. 10). Accordingly, all 12 copies of VcSiaP were in the closed state (RMSD = 0.445 over 2048 atoms of chain A to 7a5q) Interestingly, the position of VHH_VcP #2 had slightly shifted relative to its position in the VcSiaP_wt/VHH_VcP #2 complex structure (Fig. 7a, compare with Fig. 5a).

A close-up of the interaction site immediately revealed that the W73A mutant had the intended effect. The surface pocket around position 73 of VcSiaP was enlarged and F101 of CDR 3 of the VHH occupies the pocket

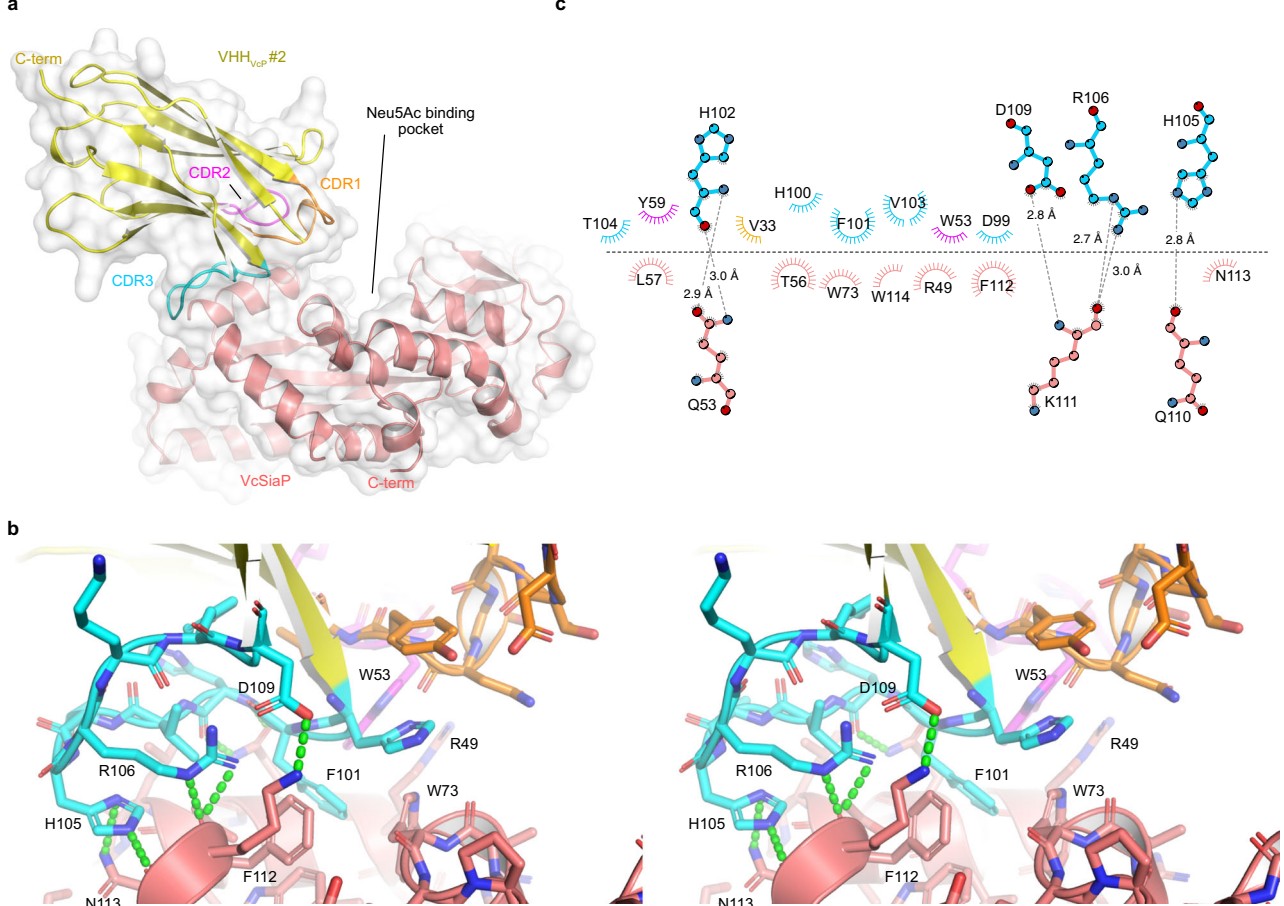

**Fig. 5 | Crystal structure of the VcSiaP/VHH_VcP #2 complex. a** Overall structure as a cartoon model. The CDR of VHH_VcP #2 are color coded throughout all panels. **b** Stereo-pair showing a close-up of the interaction. Polar interactions are indicated by green dashes. **c** Schematic representation (Ligplot+[45]) of the interacting residues (VHH_VcP #2 at the top, VcSiaP at the bottom). Polar interactions are shown as dashed lines and arcs represent van-der-Waals interactions.

without restricting the conformational change of position 73 of VcSiaP and adjacent residues.

Thus, F101 of VHH_VcP #2 acts as a steric wedge or door-stop that prevents closure of the Neu5Ac binding site of VcSiaP.

## Discussion

We isolated and characterized 11 VHH antibodies against two well studied TRAP transporter SBPs, VcSiaP and HiSiaP. For each of the two SBPs, we identified one nanobody that effectively inhibited their binding of sialic acid. The molecular reasons for this were investigated for VHH_VcP #2. Further, we analyzed the interplay between VcSiaP, VHH_VcP #2 and Neu5Ac by ITC and determined crystal structures of VHH_VcP #2 in complex with either VcSiaP or VcSiaP W73A. Since the epitope of VHH_VcP #2 is remote from the Neu5Ac binding site and because of its strongly inhibitory effect on the function of VcSiaP, the nanobody clearly acts as an allosteric inhibitor. Our data show that comparably small conformational changes within the N-terminal lobe of SiaP that normally occur during Neu5Ac binding are disturbed by F101 of VHH_VcP #2 acting like a "doorstop".

Fitting to this observation, the crystal structure of VHH_VcP #1 in complex with VcSiaP revealed that this particular VHH bound to a region that stays conformationally inert between the apo and holo states of VcSiaP, explaining why VHH_VcP #1 does not interfere with substrate binding (Fig. 6). While our results show that VHH_VcP #2 was able to effectively strip sialic acid from VcSiaP, we can at present not distinguish, whether this is due to the nanobody "prying open" the SBP or whether the nanobody simply outcompetes Neu5Ac when the SBP opens and closes[15].

Such an allosteric mechanism to control the conformational state of the SBP "from the outside" is of high interest for the study of the TRAP transporter mechanism. While it is known that the transporter preferably recognizes the closed state of the SBP[20], it is not yet clear, how exactly the formation of the tripartite complex leads to opening of the SBP and the subsequent substrate hand-over. In an AlphaFold[32] model of the tripartite complex[21], the hydrophobic loop connecting the Q3 and Q4 helices, which also contains a conserved phenylalanine, is quite close to the allosteric pocket forming the VHH_VcP #2 epitope (Fig. 8, Supplementary Fig. 11). While the exact amino acid composition of this pocket is not strongly conserved between VcSiaP, HiSiaP and other P-domains of sialic acid TRAP transporters (for instance PmSiaP)[22], similarities do clearly exist (Supplementary Fig. 11a–d). Thus, in the light of our results presented above, one might speculate that interactions between the periplasmic loops of the QM-domains and surface pockets of P domains play a role in substrate release.

Mutating residue F111 of HiSiaQ (F118 in VcSiaQ) in the center of the Q3-Q4 loop to Ala did however not show any effect in previously published uptake assays, although these in vivo experiments are not well suited to detect more subtle effects[21]. Our observation that the VcSiaP W73A mutant in the allosteric pocket has a significantly higher affinity towards Neu5Ac than the wild type protein underlines the importance of this region for the open/closed transition of the SBP. Clearly, more experimental data is needed to clarify this exciting possibility of a "push-to-release" mechanism of substrate hand over.

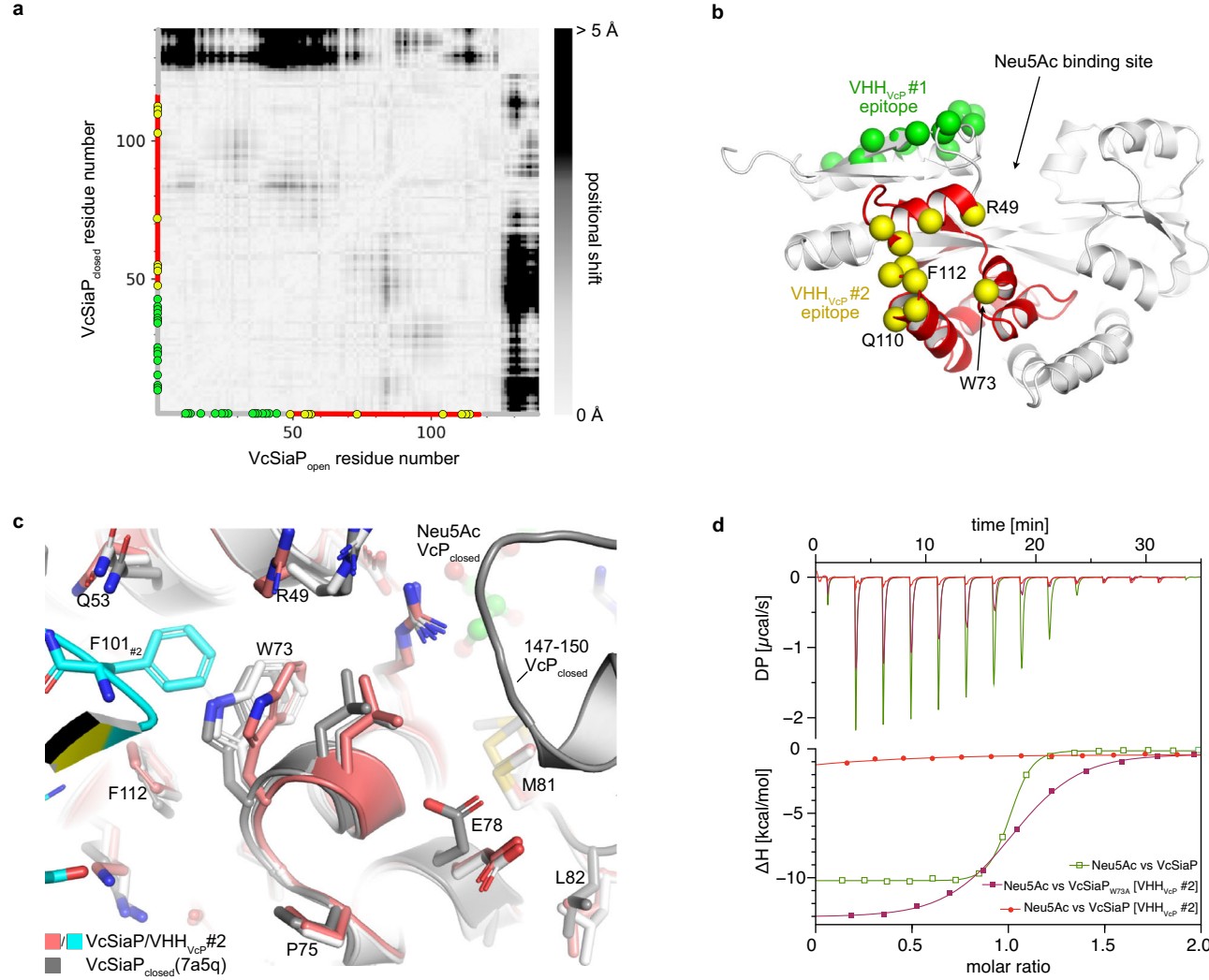

**Fig. 6 | Molecular basis for the allosteric inhibition of VcSiaP by VHH$_{VcP}$ #2. a** A difference distance matrix (calculated with mtsslSuite ([46])) of the N-terminal lobe of VcSiaP in its open- and closed-state. The white to black gradient visualizes the positional change between each pair of residues in region 1–140 of VcSiaP. Green and yellow circles indicate residues that are involved in interactions with VHH$_{VcP}$ #1 (green) or VHH$_{VcP}$ #2 (yellow). **b** A cartoon model of VcSiaP with the epitopes of VHH$_{VcP}$ #1 and VHH$_{VcP}$ #2 color coded. The region between 50 and 120 of VcSiaP that interacts with VHH$_{VcP}$ #2 is shown in red (compare to red section on axes in panel **a**)). **c** A close-up of the VcSiaP/VHH$_{VcP}$ #2 interaction in comparison to open-and closed-state VcSiaP (white and black, respectively) reveals that F101 of VHH$_{VcP}$ #2 displaces W73 of VcSiaP. **d** A set of ITC experiments revealing that VcSiaP loaded with VHH$_{VcP}$ #2 cannot bind Neu5Ac, whereas the W73A mutant of VcSiaP can bind Neu5Ac when loaded with VHH$_{VcP}$ #2. The titration of Neu5Ac to VcSiaP was performed more than $n > 5$ (biological replicates), the titration of Neu5Ac to VcSiaP + VHH$_{VcP}$ #2 (1:1) was done twice ($n = 2$, technical replicate), note that those two binding curves are also shown in Supplementary Fig. 4a. The titration of Neu5Ac to VcSiaP$_{W73A}$ + VHH$_{VcP}$ #2 (1:1) was done once ($n = 1$).

Considering that TRAP transporters play a role in the pathogenicity of for instance *H. influenzae* and *V. cholerae*[18,33–35], inhibiting the substrate uptake of SBPs is potentially a promising starting point for the development of TRAP transporter inhibitors. In light of our results, the hydrophobic pocket around W73 seems an interesting point of attack for such efforts. A possible strategy would be to find a small molecule compound that occupies this pocket and thereby mimics the effect of VHH$_{VcP}$ #2, e.g. by crystallographic fragment screening or by in silico approaches.

In conclusion, we have shown that TRAP transporter SBPs can be allosterically inhibited by VHH antibodies, another example for using these small proteins as tools to investigate biochemical processes. We found that the VHH$_{VcP}$ #2 nanobody inhibits small scale conformational changes during the open/closed transition of the SBP, explaining the allosteric effect. Our findings are an interesting starting point for efforts to block the function of VcSiaP by small molecule inhibitors.

## Materials & Methods

### Expression and purification of VcSiaP and HiSiaP

The genes for VcSiaP and HiSiaP were cloned into a pBADHisTEV vector containing a TEV cleavable N-terminal His$_6$ tag[9,36,10]. To design mutants, site directed mutagenesis was performed according to a protocol established by Liu et al.[37]. For protein purification, Ni$^{2+}$-affinity chromatography was followed by size-exclusion chromatography. The His tag was removed by incubation of a 1:50 mass ratio of TEV-protease: protein at 4 °C overnight. The purified protein was collected in the flow through of another affinity chromatography before it was concentrated, flash frozen in liquid nitrogen and stored at −80 °C.

### Discovery, expression and purification of VHHs

Within 10 weeks, one alpaca (Vicugna pacos) was immunized by six subcutaneous injections, each of 200 μg antigen (1:1 (v/v) mixture of protein solution and GERBU-FAMA adjuvant). Subsequently, peripheral blood

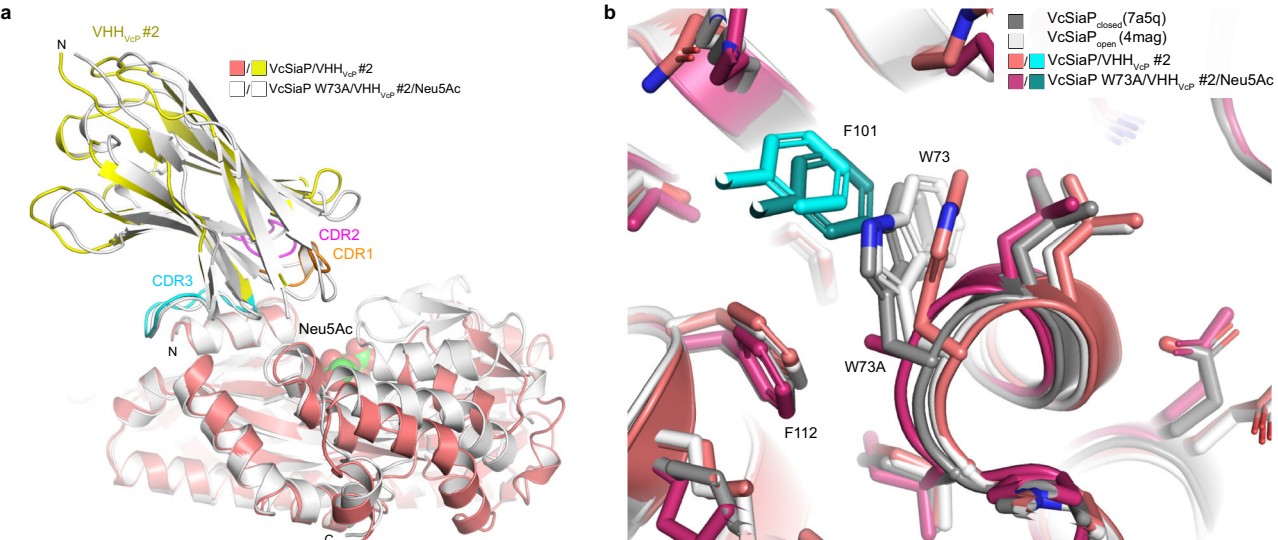

**Fig. 7 | Crystal structure of the VcSiaP W73A/VHH$_{VcP}$ #2 complex. a** An overlay of the VcSiaP W73A/VHH$_{VcP}$ #2 structure (white) with the open-state wild type complex (Fig. 5, same color coding). **b** Close-up of position 73.

mononuclear cells (PBMCs) were isolated from 100 ml of blood, their mRNA was extracted and reverse transcribed to cDNA. To generate a library for phage display, VHH sequences were amplified by PCR and cloned into a phagemid vector. The phage display was done using *E. coli TG1* cells in combination with VCSM13 helper phages to enrich specific VHHs. The biotinylated bait protein was therefore immobilized to magnetic streptavidin beads. After two rounds of panning, *E. coli ER2738* cells were infected with the enriched phages and individual clones were picked in a 96-well plate. The clones were grown 4 h at 37 °C before protein expression was induced by IPTG and nanobodies were produced over night at 30 °C. The supernatants from these small-scale expressions were tested for specific binding VHHs by ELISA. Hits were identified, sequenced and grouped according to their sequence similarity. The 'Landesuntersuchungsamt Rheinland-Pfalz' authorized all immunizations described in this study under approval number 23 177- 07/A 17-20-005 HP. We have complied with all relevant ethical regulations for animal use. Two alpacas were used: "Paco", male (*Lama glama*) 8 years; Chip number: 276094502056185 and "Zwerg" male (*Lama glama*) 7 years; Chip number: 276094502056181.

All VHHs were generated by Core Facility Nanobodies, University of Bonn. All nanobody encoding genes were provided in a pHEN6 vector with an N-terminal pelB signaling sequence and a C-terminal HA-His$_6$ tag. The plasmids were transformed into chemically competent *E. coli* WK6 cells. Cells were grown in 2 l Terific Broth (TB) media (100 µg/ml Ampicillin) inoculated with 25 ml of an overnight preculture and incubated at 37 °C and shaking until reaching an optical density of 1.2. Protein expression was induced by adding 0.4 mM IPTG followed by incubating at 37 °C and 130 rpm for 4 h. The cells were harvested by centrifugation at 4,000 r.c.f. and 10 °C for 25 min. The cell pellet was resuspended in 25 ml TES buffer (200 mM Tris, 0.65 mM EDTA, 500 mM sucrose, pH 8.0) and incubated for 1 h and slow mixing at 4 °C. For osmotic lysis, the cell suspension was diluted with 70 ml of 0.25 concentrated TES buffer and incubated at 4 °C overnight with slow mixing. Subsequently, the suspension was centrifuged at 70,000 r.c.f. and 10 °C for 45 min and the supernatant was filtered through a 0.45 µm filter. The protein solution was mixed with equilibrated Ni$^{2+}$-NTA beads and incubated for 2 h at 4 °C and slow mixing. Afterwards, Ni$^{2+}$-affinity chromatography was carried out on a gravity column. The flow through was discarded and the beads were washed with 100 ml wash buffer (50 mM Tris, 50 mM NaCl, 10 mM imidazole, pH 8.0). The protein was eluted in 15 ml elution buffer (50 mM Tris, 50 mM NaCl, 500 mM imidazole, pH 8.0) and concentrated to a final volume of 4 ml using an Amicon 3 kDa MWCO. The protein solution was loaded onto a HiLoad Superdex 75

16/600 gel filtration column and size exclusion chromatography was done in buffer A (50 mM Tris, 50 mM NaCl, pH 8.0) on an ÄKTA chromatography system. Protein containing fractions were pooled, concentrated, flash frozen in liquid nitrogen and stored at −80 °C. After each purification step, the purity was checked by SDS-PAGE.

To improve the crystallization behavior, VcSiaP specific VHHs were cloned into a pET28a vector containing an N-terminal pelB-His$_6$-TEV sequence. Therefore, the VHH encoding sequences were amplified by PCR and assembled into pET28a-pelB-His$_6$-TEV by traditional cloning using the restriction enzymes *BamH*I and *EcoR*I or *BamH*I and *Xho*I for VHH$_{VcP}$#1 and VHH$_{VcP}$#2, respectively. Briefly, the PCR products and the target vector were digested with the respective restriction enzymes and ligated at a molar ratio of 3:1 by incubation with T4 DNA ligase for 16 h at 16 °C. Positive clones were identified by double enzyme digestion and the correct sequences confirmed by sequencing at Microsynth AG (CH). Expression and purification of these constructs was done as described above. Only an additional TEV cleavage step was included to remove the His$_6$ affinity tag.

### SEC-MALS measurements
To characterize the nanobody–SBP complexes, SEC–MALS runs were performed at room temperature on an Agilent 1260 Infinity II Prime Bio LC system coupled with a Wyatt miniDAWN MALS detector, an Optilab rEX refractive index detector and a Superose6 increase 10/300 chromatography column (GE Healthcare) equilibrated with 50 mM Tris, 50 mM NaCl, pH 8.0. Data for VHH$_{HiP}$#4-VHH$_{HiP}$#11 was acquired in the same manner except for the use of a Superdex200 increase 10/300 column (GE Healthcare). All data acquisition and evaluation were carried out using ASTRA 8 software (Wyatt Technologies). The flow rate was set to 0.5 ml min$^{-1}$ and an injection volume of 50 µl was used for the experiments. The following final concentrations were used (diluted using 50 mM Tris, 50 mM NaCl, pH 8.0): 120 µM VcSiaP; 120 µM VHH$_{VcP}$#1; 120 µM VHH$_{VcP}$#2; 120 µM HiSiaP; 120 µM VHH$_{HiP}$#3; 120 µM HiSiaP; 120 µM VHH$_{HiP}$#3; 30 µM VHH$_{HiP}$#4; 30 µM VHH$_{HiP}$#5; 30 µM VHH$_{HiP}$#6; 30 µM VHH$_{HiP}$#9; 30 µM VHH$_{HiP}$#10; 30 µM VHH$_{HiP}$#11. All samples were centrifuged at 15,000 r.c.f. for 10 min before injection into the instrument.

### Isothermal Titration Calorimetry (ITC) measurements
Isothermal Titration Calorimetry with a MicroCal PEAQ-ITC (Malvern Panalytical, UK) was used to quantitatively investigate the binding behavior of VHHs towards the SBPs and Neu5Ac towards the SBPs or SBP–VHH complexes. Prior to each measurement, the sample cell was equilibrated

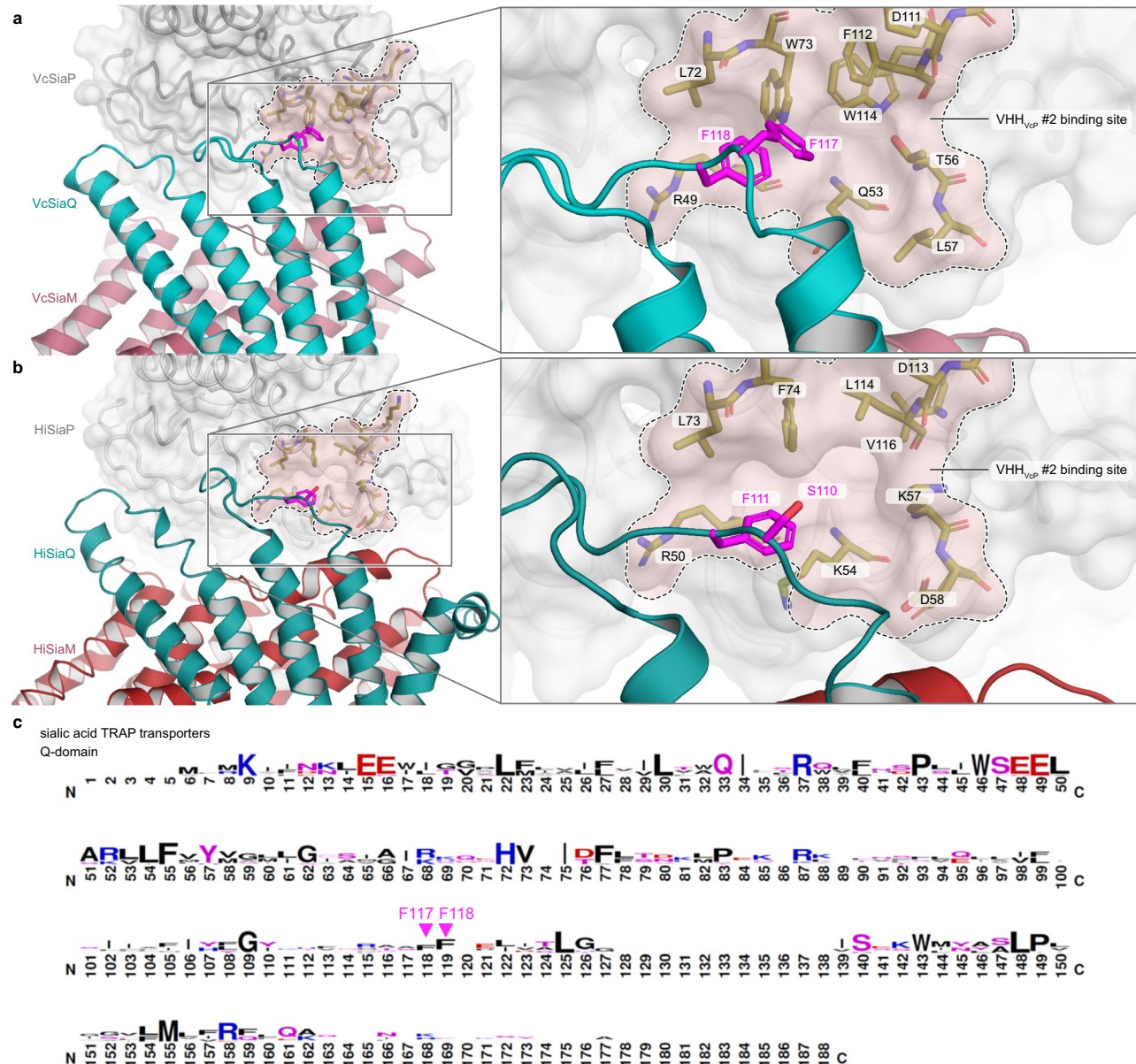

**Fig. 8 | Structural conservation of SiaPQM around the allosteric pocket that forms the VHH$_{VcP}$ #2 epitope. a** Model of the tripartite SiaPQM complex from *Vibrio cholerae*. The model was built up from AlphaFold2 predictions and experimental structures. The binding area of VHH$_{VcP}$ #2 is indicated and labelled. The Q (teal) and M(red) transmembrane domains are depicted as cartoon model and the P-domain is shown as surface representation. On the right-hand-side, a magnification of the region of interest is shown and the highlighted amino acid residues are labeled. **b** Same as (**a**) but for the tripartite SiaPQM complex from *Haemophilus influenzae*. **c** Sequence logo to visualize the hydrophobic loop of the Q-domain that is conserved among sialic acid TRAP transporters and includes the conserved Phe residue 118 (111 in HiSiaQ). This image was created with WebLogo[47] and was adapted from Peter et al.[21] (http://creativecommons.org/licenses/by/4.0/).

three times with buffer A (50 mM Tris, 50 mM NaCl, pH 8.0). Protein solution was transferred into the sample cell with a 500 μl Hamilton syringe the concentration of remaining sample solution was determined by UV absorption at λ = 280 nm with a NanoDrop 2000 (Thermo Scientific, US). The titration syringe was loaded automatically. All titration experiments were done at 25 °C, data acquisition and analysis was achieved with MicroCal PEAQ-ITC Control Software, and Analysis Software, respectively (both: Malvern Panalytical, UK). Automatic integration of the recorded thermograms yielded isotherm curves for each measurement which were fitted using the one set of site model.

To analyze the competitive effect when titrating VHH$_{VcP}$ #2 to an analyte solution of VcSiaP and Neu5Ac, a multi-step sequential ITC experiment was designed. In a first step, sialic acid was titrated to VcSiaP. The volume that was added to the initially loaded 280 μl was removed from the cell (36.4 μl). The "new" analyte concentration was estimated by using the dilution of the initial concentration ($V_0$ = 280 μl, $V_{end}$ = 316.4 μl). To estimate the concentration of the titrant in the analyte solution, the "new" analyte concentration was multiplied by the final molar ratio resulting from the titration. The syringe of the ITC device was washed and then, another titrant was loaded. In a second titration step, VHH$_{VcP}$ #2 was used as analyte. After finishing this step, the excess cell volume was again removed and "new" concentrations of the analyte contents were calculated. In a final titration step, HiSiaP was used as titrant to detect free sialic within the analyte solution.

**Dynamic Light Scattering (DLS)**
To investigate the hydrodynamic radius of the individual proteins and their complexes, protein solutions were initially prepared at a final concentration of c = 40 μM of each component. For an improved signal, the concentrations were set to or c = 114 μM in a second measurement. No significant

changes were observed upon increasing the concentration. All samples were centrifuged at 15,000 r.c.f. for 10 min to remove aggregates and measured in a DynaPro NanoStar (Wyatt Technology) DLS device using the appropriate single use cuvettes. For each condition, three measurements were done at a sample temperature of T = 25 °C and three measurement cycles of each 20 single data acquisitions with acquisition times of t = 3 s. The device was controlled and evaluation of the data was done with the DYNAMICS® software (Wyatt Technology).

### Thermal nano-DSF
Thermal denaturation curves were determined using a Prometheus NT.48 thermal nanoDSF device in combination with the PR.ThermControl software (both: NanoTemper Technologies). The sample concentration was set to 1 mg/ml of VcSiaP and potential binding partners were added at a 1.1x molar excess. All samples were centrifuged at 14,000 r.c.f. and 4 °C for 10 min and then loaded to nanoDSF grade standard glass capillaries (NanoTemper Technologies). In all experiments, samples were loaded as technical duplicates to exclude errors that can potentially occur from tiny air bubbles inside the capillary. As starting temperature $T_{start}$ = 20 °C and end temperature $T_{end}$ = 90 °C were set with a heating rate of 1 °C or 1.5 °C per minute. The different heating rates did not show effects on the observed denaturation curves.

### Crystallization and structure determination
For crystallization of 1:1 VcSiaP–VHH complexes, the protein mixtures were preincubated on ice for 30 min (590 μM VcSiaP + 750 μM VHH$_{VcP}$#1; 440 μM VcSiaP + 470 μM VHH$_{VcP}$#2). Commercial crystal screens (Molecular Dimensions, UK) were set up using a Gryphon pipetting robot (Art Robbins, US) and sitting drop crystallization plates were incubated in a Rock Imager 1000 (Formulatrix, US) crystallization hotel at 20 °C. Crystals were obtained in several conditions after two days, harvested with a cryo-loop without further cryo protection and flash frozen to liquid nitrogen. Diffraction data was recorded at beamline PX10 (Zuerich, Switzerland) with Pilatus 2 M detector at a wavelength of λ = 0.999 Å and a temperature of T = 100 K. The best individual data sets (condition Morpheus B9 for VcSiaP/VHH$_{VcP}$#1; condition Morpheus D1 for VcSiaP/VHH$_{VcP}$#2) were used for subsequent analyses.

A crystal of the VcSiaP W73A[Neu5Ac]/VHH$_{VcP}$#2 complex (405 μM VcSiaP + 2 mM Neu5Ac + 411 μM VHH$_{VcP}$#2) was obtained after 30 days of incubation at 20 °C in condition G3 of a ProPlex crystal screen (Molecular Dimensions, UK). Preparation and incubation of the crystal plates done as described above. For harvesting, the crystal was soaked with mother liquor supplemented with 35% glycerol for cryo protection. Diffraction data were collected at a wavelength of λ = 0.976 Å and a temperature of T = 100 K at beamline P13 operated by EMBL Hamburg at the PETRA III storage ring (DESY)[38].

All diffraction datasets were integrated with XDS[39]. The structures were solved by molecular replacement with PHASER[29] using PDB-IDs 4mag[7] as search model for all structures involving VcSiaP in its apo conformation or PDB-IDs 7a5q[15] as search models for the VcSiaP W73A-closed VHH_VcP_#2 structure. For initial nanobody structures, a BtuF specific VHH (PDB ID: 5ovw_g[30]) was used as search model, and the sequence was adapted by hand. These obtained structures were used as search models for the VcSiaP W73A/Neu5Ac/VHH$_{VcP}$#2 structure. The structure refinement processes were achieved with phenix.refine[40], COOT[41] and ISOLDE[42]. After each refinement step, MolProbity[43] was used to check the model quality. The Ramachandran statistics were determined for all three structures. For the VcSiaP+VHH$_{VcP}$#1 structure, 97.04% of all residues were found in favored regions, 2.61% in allowed regions, and 0.36% are considered as outliers. For the VcSiaP+VHH$_{VcP}$#2 structure, 97.85% were found in favored regions, 2.03% in allowed regions, and 0.12% are considered as outliers. For the VcSiaP W73A+Neu5Ac+VHH$_{VcP}$#2 structure, 97.37% of all residues were found in favored regions, 1.89% in allowed regions, and 3.02% are considered as outliers. All figures were prepared in PyMOL (www.pymol.org).

### Statistics and reproducibility
Information concerning statistics and reproducibility for the experiments shown in this study are given in the figure legends of the corresponding experiments. If multiple experiments were performed, the number of individual experiments (n), their average value and the standard deviation are given.

### Reporting summary
Further information on research design is available in the Nature Portfolio Reporting Summary linked to this article.

### Data availability
The coordinates and structure factors of the crystal structures determined in this study have been deposited to the PDB under accession codes 9fvc, 9fvb and 9fve. Structures with the following PDB accession codes were used for discussion of our results: 7a5q, 3b50, 7qe5, 7t3e, 7qha, 4mmp. The graph source data can be found in Supplementary Data 1. All other data are available from the corresponding author on reasonable request.

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

## Acknowledgements
GH is grateful for funding by the DFG (HA6805/5-1 and HA6805/5-2). Part of this work was funded by a Method Development Grant of the TRA Life and Health (University of Bonn) as part of the Excellence Strategy of the federal and state governments. The synchrotron data was collected at beamline PX10 of the swiss light source (SLS) (Zuerich, Switzerland) and at beamline P13 operated by EMBL Hamburg at the PETRA III storage ring (DESY, Hamburg, Germany). We would like to thank Matthias Geyer (Institute of Structural Biology, University of Bonn) for support and discussions.

## Author contributions
N.S. expressed and purified all VcSiaP, VHH$_{VcP}$ #1 and VHH$_{VcP}$ #2 constructs, performed the crystallization and biochemical analyses. P.H. did the panning and cloning for VHH$_{HiP}$ #4-11 and performed the SEC-MALS and ITC experiments for those VHHs. N.S. and P.H. expressed and purified VHH$_{HiP}$ #4-11. E.G. expressed and purified VHH$_{HiP}$ #3 and investigated its binding behavior to VcSiaP and HiSiaP. S.B. cloned the expression vectors for VHH$_{VcP}$ #1 and VHH$_{VcP}$ #2 with N-terminal, cleavable His$_6$ tag. N.S. and M.F.P. designed all mutants used in this work. S.M. and P.A.K. supervised the generation of VHH antibodies. N.S., P.H., M.F.P., G.H.T., S.M. and G.H. analyzed data. N.S. and G.H. wrote the manuscript with input from all authors. G.H. supervised the study and acquired funding.

## Funding

## Competing interests
The authors declare no competing interests.
