## [Peer Review file · Communications Biology]

Allosteric substrate release by a sialic acid TRAP transporter substrate binding protein

Corresponding Author: Dr Gregor Hagelueken

Version 0:

Reviewer comments:

Reviewer #1

(Remarks to the Author)

Summary of the manuscript:

The authors have raised nanobodies against two SiaPs (one from Hi and another from Vc). The SiaPs are interesting drug targets, and hence, the study provides important insights into the mechanism of substrate binding - that will help in drug discovery. They identified nanobodies that bind to VcSiaP and HiSiaP and have characterized the binding affinities by ITC and, if more than one nanobody can bind simultaneously to these proteins by SEC-MALS. They make an interesting observation of one of the VcSiaP nanobodies - in its ability to bind to the VcSiaP-Neu5Ac complex and disassociate Neu5Ac from binding to it. Combined with the observation that if the VcSiaP is already bound to the nanobody, it does not bind to Neu5Ac, they propose that this is an allosteric binding. They provide further evidence by X-ray crystallographic structure determinations.

Overall impression:

The work is very thorough. The studies and the data are robust. The only challenge I have is that I do not think this characterizes the binding as allosteric. Should this be instead called noncompetitive binding? Allosterity, as a definition, modulates function. To demonstrate allosterity, one should demonstrate a change in the Hill Coefficient. In the absence of a molecule binding to a different site, inhibits substrate binding at the active site, it should be noncompetitive inhibition. If the Nanobody binds to the SiaP-Neu5Ac complex and stops transfer to SiaT, then it would be an uncompetitive inhibitor. Given the fact that these have not been tested - I would accept the the definition as an noncompetitive inhibitor. Allosterity is a stretch.

But, otherwise, the paper is well argued, and I support it for publication.

Specific comments:

Major:

1. Figures 8 and 380 to 400 are not really relevant to the study and can be deleted. Supplementary Figure 11 is also not relevant in the same way.

2. Around Line 150: It would be nice to see the curves. These curves should so multi-stage denaturation curves - like multi-domain proteins. If they show a single sigmoidal curve, then are they observing the denaturation of SiaP or the nanobody? Instead of just showing melting temperatures as bar charts, that would be more useful.

Minor:

Line 355: Should you add a 3rd possibility that the binding could stabilize the open conformation, and this is not true in the W73A mutation?

Line 440 (Extended Data Table 2): Can we add here also the data for the SiaP-W73A? That information is critical.

I know this is also about getting crystals - but the structure of SiaP[VHHVCP#1] with Neu5Ac - Where Neu5Ac is added after the complex is formed - will show the link between kinetics and thermodynamics..

Supplementary Figure 1: Am I seeing things wrong? In both cases, the calculated N is slightly less than 1. But from the graphs, my intuition tells me it is greater than 1.00. What am I seeing wrong?

It is good practice to subtract the heat of dilution (which is the DP that you get at the end of binding) from all data. I do not think the values will change very much, so I am OK with it if they do not.

Can we rephrase Lines 185 - 195? I had to read them several times to understand them. There is nothing wrong with them, but they were really difficult to understand.

Reviewer #2

(Remarks to the Author)

Dear Editor,
Dear Authors

Thank you for the invitation to revise the manuscript by Schneberger et al. In this manuscript, the authors perform selection of nanobodies targeting the substrate binding protein (SBP), SiaP. The function of this protein is to bind and deliver sialic acid to a tripartite ATP-independent periplasmic (TRAP) transporter system, which will result in import of sialic acid into the cell.

The manuscript presents a compelling study where two nanobodies are shown to inhibit the binding of sialic acid to SiaP. The authors elucidate the nanobody-bound crystal structures and employ ITC and SEC-MALS to characterize the interactions between these proteins. The findings suggest that sialic acid binding is abolished through an allosteric mechanism involving hydrophobic residues on the surface of SiaP.

Overall, the research questions addressed in this manuscript are significant for advancing our understanding of the molecular mechanism of transport by TRAP transporters. In principle, I would support the publication of this manuscript after the authors address the following concerns:

1. Impact on transport: Although the structural and functional characterization of the binding of nanobodies 1 and 2 to SiaP are robust, the manuscript could benefit from exploring whether this binding influences the transport of sialic acid by the TRAP transporter. Addressing this question would substantially enhance the relevance of this study, particularly given the importance of TRAP transporters in pathogenic bacteria.

2. Broader impact of the proposed model: In lines 375-385, the authors propose a model, based on crystal structures, mutagenesis analysis, and AlphaFold modeling, suggesting that the TRAP transporter might influence the conformational state of SiaP. It would be beneficial if the authors could elaborate on this model within a broader context, perhaps by comparing it to other TRAP transporters or similar substrate-binding protein (SBP) systems.

Reviewer #3

(Remarks to the Author)

The authors generated and extensively characterized eleven nanobodies (camelid VHH antibodies) directed against periplasmic binding proteins (called SiaP) of tripartite ATP-independent periplasmic (TRAP) transporters for sialic acid from two important human pathogens, namely *Haemophilus influenzae* and *Vibrio cholerae*. For the nanobodies against *H. influenzae* they determined affinity by ITC and mapped common epitopes. For two nanobodies against *V. cholerae* they additionally determined crystal structures in complex with SiaP. These structures provide an elegant mechanistic explanation how one of the two nanobodies allosterically inhibits binding of sialic acid. The authors suggest that this knowledge could be used to design small molecule drugs to prevent binding of sialic acid to SiaP.

This is a beautiful paper. It deals with a medically important question and presents a mechanistically interesting answer. The paper summarizes a huge amount of work. The analysis is expertly performed. The text is succinct and clear. The figures are very well designed and illustrate the points that the authors want to make. Overall, I have no criticism but only some questions and suggestions that may help to improve a tiny bit further this already wonderful manuscript.

Specific minor points:

- 1) Line 48: Gram-positive bacteria have only one membrane. Does it make sense to call this "inner membrane"?
- 2) Line 114: "chromatography" is missing after "size exclusion"
- 3) Line 143-145: The authors could (or maybe should) mention their SEC-MALS results shown in Fig. 2A for VcSiaP + #1 + #2. I consider this even more convincing than the DLS mentioned in the text.
- 4) Line 272: How was the DDM calculated?
- 5) Fig. 6d: Why is $-\Delta H$ larger (ΔH more negative) for W73A [VHHVCP #2] than for the wt in the lower panel, although the peaks in the upper panel are less negative (and apparently of a very similar width) for the mutant in the upper panel?
- 6) Line 298: Why use "albeit"? To me, a contrasting conjunction does not seem to fit here.

- 7) Line 347: I do not really consider crystal structures between 2.0 and 2.8 Å to be high resolution.
- 8) Line 410-419: No author contribution is listed for PAK.
- 9) Extended Data Table 1: The header should mention the method to determine experimental molecular weights.
- 10) Extended Data Table 2: It would be nice to get more details on the competitive model for 2). Is there an equation one could state?
- 11) Extended Data Table 3: Space group names should be set according to convention (P and C in italics, 1 in 21 screw axis as subscript).
- 12) Suppl. Fig. 1 b: It seems to me that two measurement points are missing in the lower panel (between first and second and between third and fourth red dot). Is there any reason for this?
- 13) Suppl. Fig. 2 b: The MW curve from MALS is missing for #9,#10,#11.
- 14) Suppl. Fig. 2 a/b: Do the authors have any idea why VHHs show very different elution volumes ranging from about 17.2 to 19.4 ml? Do they tend to dimerize in the absence of the antigen? Or do they interact with the matrix?
- 15) Suppl. Fig. 2 panel d: The description of panel d) is missing in the legend.
- 16) Suppl. Fig. 2 panel d): Why does HiSiaP have a different elution volume than in panels a) and b)? Is this a different column? There seems to be only a very small shift from HiSiaP to HiSiaP [VHH-HiP #3]. Why?
- 17) Suppl. Fig. 13 b: What do the individual dots show?
- 18) PDB validation reports: All reports state "EDS failed to run properly". Thus, all data-based metrics cannot be judged based on the validation reports. The EDS failure might be a problem on the PDB server side. More likely, this indicates some problem(s) with the deposited structure factors. I consider the map coefficients provided by PDB for all structures very useful. Hence, I encourage the authors to check what the problem is. If they can locate the problem, I would encourage them to re-deposit usable structure factors.

Version 1:

Reviewer comments:

Reviewer #1

(Remarks to the Author)

I have read the responses and the revised manuscript. I support publication of the manuscript.

Reviewer #2

(Remarks to the Author)

The authors have addressed my previous concerns and adequately incorporated the new data into their analysis. Based on this, I recommend the publication of this manuscript.

Reviewer #3

(Remarks to the Author)

The authors have responded satisfactorily to all my comments.

There is one potential typo in line 239 that I forgot to mention in my first review. Q99 should probably be E99 as in Fig. 4. This comment does not require another round of review and I recommend publication of the manuscript.

REVIEWERS' COMMENTS:

We would like to thank all three referees again for their time and effort in reviewing our work!

Reviewer #1 (Remarks to the Author):

I have read the responses and the revised manuscript. I support publication of the manuscript.

Thank you for supporting our revised manuscript for publication.

Reviewer #2 (Remarks to the Author):

The authors have addressed my previous concerns and adequately incorporated the new data into their analysis. Based on this, I recommend the publication of this manuscript.

Thank you for reviewing our revised manuscript and recommending it for publication.

Reviewer #3 (Remarks to the Author):

The authors have responded satisfactorily to all my comments.

There is one potential typo in line 239 that I forgot to mention in my first review. Q99 should probably be E99 as in Fig. 4.

Thanks, we corrected this typo. Furthermore, we corrected the bond lengths included in the text. They now match the values given in the figure.

This comment does not require another round of review and I recommend publication of the manuscript.

Thank you for your constructive comments during the review process and for recommending the revised manuscript for publication.

Dear Dr Hagelueken,

Your manuscript entitled "Allosteric substrate release by sialic acid TRAP transporter SBPs" has now been seen by 3 referees. You will see from their comments below that while they find your work of considerable interest, some important points are raised. We are interested in the possibility of publishing your study in Communications Biology, but would like to consider your response to these concerns in the form of a revised manuscript before we make a final decision on publication.

We therefore invite you to revise and resubmit your manuscript, taking into account the points raised. In particular,

-Please consider the comment from R1 about allosteric/non competitive binding, and please provide the corresponding curves.

You will see that we have taken this comment of Referee#1 very serious and have provided the plots that the referee asked for. However, we are convinced that our definition of the Nanobody as an allosteric inhibitor is correct. We hope that our answer will convince both you and Referee#1.

Please highlight all changes in the manuscript text file.

We would like to thank all three reviewers for their time and effort and their very favorable comments.

Reviewers' comments:

Reviewer #1 (Remarks to the Author):

Summary of the manuscript:

The authors have raised nanobodies against two SiaPs (one from Hi and another from Vc). The SiaPs are interesting drug targets, and hence, the study provides important insights into the mechanism of substrate binding - that will help in drug discovery. They identified nanobodies that bind to VcSiaP and HiSiaP and have characterized the binding affinities by ITC and, if more than one nanobody can bind simultaneously to these proteins by SEC-MALS. They make an interesting observation of one of the VcSiaP nanobodies - in its ability to bind to the VcSiaP-Neu5Ac complex and disassociate Neu5Ac from binding to it. Combined with the observation that if the VcSiaP is already bound to the nanobody, it does not bind to Neu5Ac, they propose that this is an allosteric binding. They provide further evidence by X-ray crystallographic structure determinations.

Overall impression:

The work is very thorough. The studies and the data are robust. The only challenge I have is that I do not think this characterizes the binding as allosteric.

Thank you very much for your very detailed and favorable comments. Your comment about our use of the word allostery has prompted us to refresh our knowledge about the definition of this term.

The concept of allostery was first introduced in the 1960s to describe the mechanism of feedback inhibition exerted on bacterial enzymes by regulatory ligands. It was used by Monod and colleagues, to explain the complex kinetics of haemoglobin. (Changeux, J.-P. Allostery and the Monod-Wyman-Changeux Model After 50 Years. *Annu. Rev. Biophys.* **41**, 103–133 (2012).)

The term is still widely used in situations where a molecule binds to a distinct site on a protein and modulates the protein's function in a positive or negative way. The term is not only used in the case of enzymes but also for receptors and, as in our case, substrate binding proteins (SBPs).

We would also like to note that within our field there is precedence from at least one important study, where antibodies have been shown to bind to SBPs at sites away from the binding cavity and modulate the affinity of the protein for its cognate ligand. (Rizk, S. S. *et al.* Allosteric control of ligand-binding affinity using engineered conformation-specific effector proteins. *Nat. Struct. Mol. Biol.* **18**, 437–442 (2011)). The authors of this study isolated antibodies that modulated the affinity of a substrate binding protein, here the maltose binding protein (MBP), to have increased affinity for the sugar. Based on the above and the paper by Rizk *et al.*, we would argue that the term allostery is in our case not inappropriate.

Should this be instead called noncompetitive binding?
Allostery, as a definition, modulates function.

Clearly, the Nanobody does not merely bind to the SBP but has a drastic effect on its function, namely to bind its substrate sialic acid.

To demonstrate allostery, one should demonstrate a change in the Hill Coefficient.

In our manuscript, we show ITC titrations of VcSiaP with Neu5Ac in the presence of three different concentrations of the nanobody. We plotted this data as Hill-plots by calculating the free ligand concentration after each injection. This is possible, since we know the exact volume and the amounts of SBP and ligand, as well as the K_D and the fraction of Neu5Ac-bound to SBP (by dividing the heat exchanged up to this injection by the total heat exchanged). These plots were then fitted with the Hill equation.

In the absence of Nb, the fitted Hill coefficient is $n=1.3$ and thus close to 1.0, as expected for this well-known system. For the second plot (0.5 molar equivalents of Nb), the fraction bound decreases to ~ 0.5 , because the stronger binding nanobody blocks half of the SBPs. The Hill coefficient does not change significantly ($n=1.5$). In the presence of 1.0 molar equivalents of nanobody, no binding is observed and we cannot calculate a Hill coefficient.

Please also note that it is disputed, whether allostery can be detected by analyzing Hill coefficients (Prinz, H. Hill coefficients, dose-response curves and allosteric mechanisms. *J. Chem. Biol.* 3, 37–44 (2010)).

In the absence of a molecule binding to a different site, inhibits substrate binding at the active site, it should be noncompetitive inhibition. If the Nanobody binds to the SiaP-Neu5Ac complex and stops transfer to SiaT, then it would be an uncompetitive inhibitor. Given the fact that these have not been tested - I would accept the the definition as an noncompetitive inhibitor. Allostery is a stretch.

There are at least four different inhibitory mechanisms that are commonly discussed (“competitive”, “uncompetitive”, “non-competitive” and “mixed”). But, the crucial question here is: How is the inhibitory effect achieved? The referees’ example with a nanobody blocking the transfer from the SBP to the transmembrane domains can clearly be achieved without affecting the binding of the substrate to the SBP itself. This basically resembles the case of VcSiaP/VHH_{VCP} #1.

In contrast, the inhibitory effect of VHH_{VCP}#2 is achieved by the nanobody binding to an allosteric site (per definition a distinct site, away from the “active site”) and directly affecting the prime function of the SBP by changing its 3D structure (see above).

Taken together, we think that VHH_{VCP}#2 is very clearly an allosteric inhibitor of VcSiaP. We have now included a short explanation of why we use the term allosteric inhibitor in the discussion section.

But, otherwise, the paper is well argued, and I support it for publication.

Thank you very much.

Specific comments:

Major:

1. Figures 8 and 380 to 400 are not really relevant to the study and can be deleted. Supplementary Figure 11 is also not relevant in the same way.

We think that the discussion section of our paper is a good place to think about a possible mechanistic relevance of our findings, especially because the question of how the SBP is opened during substrate handover is still completely unclear. We think that such hypotheses (we have marked it as such) are of high interest and would like to keep this section and Supplementary Figure 11.

Please note that referee #2 specifically asked us to elaborate on this point.

2. Around Line 150: It would be nice to see the curves. These curves should show multi-stage denaturation curves - like multi-domain proteins. If they show a single sigmoidal curve, then are they observing the denaturation of SiaP or the nanobody? Instead of just showing melting temperatures as bar charts, that would be more useful.

We now provide the curves in supplementary Figure 3 d. However, we think the bar diagram is more convenient to get a fast overview and therefore we have also left the bar diagram in the paper. We observed only one melting point for the different protein complexes, which differed significantly from the melting temperatures of the individual proteins. We have observed a similar behaviour in other nanoDSF experiments with completely different VHH protein complexes. (see: Kopp, A., Hagelueken, G., Jamitzky, I. et al. Pyroptosis-inhibiting nanobodies block Gasdermin D pore formation. *Nat Commun* 14, 7923 (2023). <https://doi.org/10.1038/s41467-023-43707-z>). Multi-step denaturation curves were observed when a binding partner was used in excess, showing one melting point for the complex and another for the single protein at higher concentration.

Minor:

Line 355: Should you add a 3rd possibility that the binding could stabilize the open conformation, and this is not true in the W73A mutation?

This possibility is actually meant by "the nanobody simply outcompetes Neu5Ac when the SBP opens and closes" (line 357-358). The open state stabilization is not given using the W73A mutant, which is shown by the W73A mutant crystal structure bound to VHH_{VCP#2} and Neu5Ac. And is also supported by the *in vitro* data we showed for that mutant.

Line 440 (Extended Data Table 2): Can we add here also the data for the SiaP-W73A? That information is critical.

Agreed! The data for the W73A mutant is now included in extended data table 2.

I know this is also about getting crystals - but the structure of SiaP[VHHVCP#1] with Neu5Ac - Where Neu5Ac is added after the complex is formed - will show the link between kinetics and thermodynamics..

Do you mean SiaP[VHH_{VCP#2}] with Neu5Ac? We indeed tried to get crystals, but unfortunately without success.

Supplementary Figure 1: Am I seeing things wrong? In both cases, the calculated N is slightly less than 1. But from the graphs, my intuition tells me it is greater than 1.00. What am I seeing wrong?

You're right. However, the given values are averaged from 3 individual measurements. The graph is showing only one of these experiments. This is now stated in the figure legend.

It is good practice to subtract the heat of dilution (which is the DP that you get at the end of binding) from all data. I do not think the values will change very much, so I am OK with it if they do not.

Thanks for the comment. As you mentioned, there is no large dilution effect of the buffer because we use exactly the same in both cell and syringe. We therefore decided to leave it as it is.

Can we rephrase Lines 185 - 195? I had to read them several times to understand them. There is nothing wrong with them, but they were really difficult to understand.

We tried to make it easier to understand by shortening the sentences.

Reviewer #2 (Remarks to the Author):

Dear Editor,
Dear Authors

Thank you for the invitation to revise the manuscript by Schneberger et al. In this manuscript, the authors perform selection of nanobodies targeting the substrate binding protein (SBP), SiaP. The function of this protein is to bind

and deliver sialic acid to a tripartite ATP-independent periplasmic (TRAP) transporter system, which will result in import of sialic acid into the cell.

The manuscript presents a compelling study where two nanobodies are shown to inhibit the binding of sialic acid to SiaP. The authors elucidate the nanobody-bound crystal structures and employ ITC and SEC-MALS to characterize the interactions between these proteins. The findings suggest that sialic acid binding is abolished through an allosteric mechanism involving hydrophobic residues on the surface of SiaP.

Overall, the research questions addressed in this manuscript are significant for advancing our understanding of the molecular mechanism of transport by TRAP transporters. In principle, I would support the publication of this manuscript after the authors address the following concerns:

1. Impact on transport: Although the structural and functional characterization of the binding of nanobodies 1 and 2 to SiaP are robust, the manuscript could benefit from exploring whether this binding influences the transport of sialic acid by the TRAP transporter. Addressing this question would substantially enhance the relevance of this study, particularly given the importance of TRAP transporters in pathogenic bacteria.

You're right! We plan to do such experiments for a detailed follow up study.

2. Broader impact of the proposed model: In lines 375-385, the authors propose a model, based on crystal structures, mutagenesis analysis, and AlphaFold modeling, suggesting that the TRAP transporter might influence the conformational state of SiaP. It would be beneficial if the authors could elaborate on this model within a broader context, perhaps by comparing it to other TRAP transporters or similar substrate-binding protein (SBP) systems.

Please note that we do show further models in Supplementary figure 11 and discuss them in lines 390 and below.

Reviewer #3 (Remarks to the Author):

The authors generated and extensively characterized eleven nanobodies (camelid VHH antibodies) directed against periplasmic binding proteins (called SiaP) of tripartite ATP-independent periplasmic (TRAP) transporters for sialic acid from two important human pathogens, namely *Haemophilus influenzae* and *Vibrio cholerae*. For the nanobodies against *H. influenzae* they determined affinity by ITC and mapped common epitopes. For two nanobodies against *V. cholerae* they additionally determined crystal structures in complex with SiaP. These structures provide an elegant mechanistic explanation how one of the two nanobodies allosterically inhibits binding of sialic acid. The authors suggest that this knowledge could be used to design small molecule drugs to prevent binding of sialic acid to SiaP.

This is a beautiful paper. It deals with a medically important question and presents a mechanistically interesting answer. The paper summarizes a huge amount of work. The analysis is expertly performed. The text is succinct and clear. The figures are very well designed and illustrate the points that the authors want to make. Overall, I have no criticism but only some questions and suggestions that may help to improve a tiny bit further this already wonderful manuscript.

Thank You very much!

Specific minor points:

1) Line 48: Gram-positive bacteria have only one membrane. Does it make sense to call this "inner membrane"?
Agreed, we have changed it to just "membrane".

2) Line 114: "chromatography" is missing after "size exclusion"

Thanks, we corrected this.

3) Line 143-145: The authors could (or maybe should) mention their SEC-MALS results shown in Fig. 2A for VcSiaP + #1 + #2. I consider this even more convincing than the DLS mentioned in the text.

We now mentioned this experiment and included the link to the respective figure panels.

4) Line 272: How was the DDM calculated?

We used the mtsslSuite (www.mtsslsuite.isb.ukbonn.de) and have added "Hagelueken et. al" as reference to the figure caption

5) Fig. 6d: Why is $-\Delta H$ larger (ΔH more negative) for W73A [VHHVCP #2] than for the wt in the lower panel, although the peaks in the upper panel are less negative (and apparently of a very similar width) for the

mutant in the upper panel?

Different concentrations were used in the experiments (50µM VcSiaP W73A: 500µM Neu5Ac; 100µM VcSiaP WT: 1200µM Neu5Ac). Higher concentration of course leads to a larger DP, whereas the number in the bottom panel refer to molar deltaH and therefore account for the different concentrations used.

6) Line 298: Why use "albeit"? To me, a contrasting conjunction does not seem to fit here.
You're right, we deleted the word "albeit".

7) Line 347: I do not really consider crystal structures between 2.0 and 2.8 Å to be high resolution.
Ok, we changed this to only "crystal structures"

8) Line 410-419: No author contribution is listed for PAK.
Thanks, we amended this.

9) Extended Data Table 1: The header should mention the method to determine experimental molecular weights.
Ok, it's included now.

10) Extended Data Table 2: It would be nice to get more details on the competitive model for 2). Is there an equation one could state? We used the ITC analysis software by Malvern Panalytical to analyse all ITC data. For the competitive models, the manual references the following paper which includes all equations (Sigurskjold, B. W. Exact Analysis of Competition Ligand Binding by Displacement Isothermal Titration Calorimetry. Anal. Biochem. 277, 260–266 (2000).). The reference has been added to the paper.

11) Extended Data Table 3: Space group names should be set according to convention (P and C in italics, 1 in 21 screw axis as subscript).
Ok, we changed this, also for Extended Data Table 4.

12) Suppl. Fig. 1 b: It seems to me that two measurement points are missing in the lower panel (between first and second and between third and fourth red dot). Is there any reason for this?
These data points were excluded for the calculation of the binding curve because the injection signals were overlaid with some artefacts caused by the injection. We now included the data points in the figure but treated them as outliers and did not include them in the fitting calculations. This explanation has been added to the figure legend.

13) Suppl. Fig. 2 b: The MW curve from MALS is missing for #9,#10,#11.
The MALS curves for the individual measurements of the VHHs alone are not shown because these curves could not be recorded properly. However, we included the UV chromatograms of the measurements, as they provide a good control at which position an unbound VHH would be observed for the SEC-MALS runs of the complexes. We mentioned this in the figure caption.

14) Suppl. Fig. 2 a/b: Do the authors have any idea why VHHs show very different elution volumes ranging from about 17.2 to 19.4 ml? Do they tend to dimerize in the absence of the antigen? Or do they interact with the matrix?
In addition to the referee's suggestion we can only speculate that this might be due to the differing lengths of CDR loops, the hydrodynamic radii differ and thus the elution volume is affected.

15) Suppl. Fig. 2 panel d: The description of panel d) is missing in the legend.
Thanks, it is included now.

16) Suppl. Fig. 2 panel d): Why does HiSiaP have a different elution volume than in panels a) and b)? Is this a different column? There seems to be only a very small shift from HiSiaP to HiSiaP [VHH-HiP #3]. Why?
We included the column type in the figure to highlight that different columns were used for the experiments shown in a) and b) (SD200 increase 10/300) and that shown in d) (Superose6 increase 10/300). The small, but observable peak shift can be explained by a much weaker affinity for VHH-HiP#3 compared to all other VHHs.

17) Suppl. Fig. 13 b: What do the individual dots show?

We assume you mean Suppl. Fig. 3b. Each dot corresponds to one individual measurement which includes 20 single acquisitions á 3 seconds. This is now explained in the legend.

18) PDB validation reports: All reports state “EDS failed to run properly”. Thus, all data-based metrics cannot be judged based on the validation reports. The EDS failure might be a problem on the PDB server side. More likely, this indicates some problem(s) with the deposited structure factors. I consider the map coefficients provided by PDB for all structures very useful. Hence, I encourage the authors to check what the problem is. If they can locate the problem, I would encourage them to re-deposit usable structure factors.

This was indeed an error by the PDB. The EDS ran without problem in the initial validation report (the one that is marked “not for manuscript review”), but failed in the full validation report (the one marked “for manuscript review”). We had submitted both reports to the editor. The PDB staff has now corrected the problem and the error no longer appears in the full validation report.